# Antibiotic Consumption Patterns in European Countries Are Associated with the Prevalence of Parkinson’s Disease; the Possible Augmenting Role of the Narrow-Spectrum Penicillin

**DOI:** 10.3390/antibiotics11091145

**Published:** 2022-08-23

**Authors:** Gábor Ternák, Márton Németh, Martin Rozanovic, Gergely Márovics, Lajos Bogár

**Affiliations:** 1Institute of Migration Health, Medical School, University of Pécs, Szigeti Str. 12., H-7624 Pécs, Hungary; 2Department of Anesthesiology and Intensive Care, Medical School, University of Pécs, Szigeti Str. 12., H-7624 Pécs, Hungary; 3Department of Public Health Medicine, Medical School, University of Pécs, Szigeti út 12., H-7624 Pécs, Hungary

**Keywords:** Parkinson’s disease (PD), microbiome, dysbiosis, antibiotics, antibiotic consumption, narrow-spectrum penicillin, Lewy-bodies (LB), neurotransmitter, substantia nigra, curli protein

## Abstract

Parkinson’s disease: Parkinson’s disease (PD) is the second-most common neurodegenerative disease, affecting at least 0.3% of the worldwide population and over 3% of those over 80 years old. According to recent research (2018), in 2016, 6.1 million (95% uncertainty interval (UI) 5.0–7.3) individuals had Parkinson’s disease globally, compared with 2.5 million (2.0–3.0) in 1990. The pandemic-like spreading of PD is considered a slow-moving disaster. Most recent studies indicated the possible role of an altered microbiome, dysbiosis, in the development of PD, which occurs long before the clinical diagnosis of PD. Antibiotics are considered as major disruptors of the intestinal flora and we have hypothesized that, as different classes of antibiotics might induce different dysbiosis, certain classes of antibiotics could trigger the PD-related dysbiosis as well. Comparative analyses were performed between the average yearly antibiotic consumption of 30 European countries (1997–2016) and the PD prevalence database (estimated for 2016). We divided the time frame of antibiotic consumption of 1997–2016 into four subsections to estimate the possible time lapse between antibiotic exposure and the prevalence, prevalence change, and PD-related death rates estimated for 2016. Our results indicated that countries with high consumption of narrow-spectrum penicillin experienced a higher increase in PD prevalence than the others. Countries reporting a decline in PD from 1990 to 2016 demonstrated a reduction in the consumption of narrow-spectrum penicillin in this period.

## 1. Introduction

According to recent reports [1], in 2016, 6.1 million (95% uncertainty interval (UI) 5.0–7.3) individuals had Parkinson’s disease globally, compared with 2.5 million (2.0–3.0) in 1990.

As of the WHO report (13 June 2022), globally, disability and death due to PD are increasing faster than for any other neurological disorder. The prevalence of PD has doubled in the past 25 years. Global estimates in 2019 showed over 8.5 million individuals with PD. Current estimates suggest that, in 2019, PD resulted in 5.8 million disability-adjusted life years, an increase of 81% since 2000, and caused 329,000 deaths, an increase of over 100% since 2000 (https://www.who.int/news-room/fact-sheets/detail/parkinson-disease, accessed on 18 August 2022). The pandemic-like spreading of PD is considered a slow-moving disaster. Parkinson’s disease occurs when nerve cells, or neurons, in the brain die or become impaired. Although many brain areas are affected, the most common symptoms result from the loss of neurons in an area near the base of the brain called the substantia nigra [2]. Normally, the neurons in this area produce an important brain chemical known as dopamine. Dopamine is a chemical messenger responsible for transmitting signals between the substantia nigra and the next “relay station” of the brain, the corpus striatum, to produce smooth, purposeful movement. Loss of dopamine results in abnormal nerve-firing patterns within the brain that cause impaired movement. Studies have shown that most people with Parkinson’s have lost 60 to 80 percent or more of the dopamine-producing cells in the substantia nigra by the time symptoms appear and that people with PD also have a loss of the nerve endings that produce the neurotransmitter norepinephrine [3,4].

The pathological hallmark of PD is the accumulation of filamentous, cytoplasmic inclusions consisting mainly of α-synuclein aggregations in the form of Lewy bodies (LB) or Lewy neurites (LN). α-Synuclein phosphorylation and fibrillization lead to LB formation and induce neuron death [5,6]. LB are found in certain areas of the CNS, e.g., the basal ganglia, the dorsal motor nucleus of the vagus (NV), the olfactory bulb (OB), the locus coeruleus (LC) and the intermediolateral nucleus in the spinal cord (IML); and of the peripheral nervous system (PNS), e.g., the celiac ganglia and enteric nervous system (ENS) of PD patients [7,8,9]. Several etiological factors were implicated as causative agents promoting the development of PD, out of which genetics, toxic agents (such as toxic fumes, manganese, etc.), and mitochondrial injury could be mentioned [10].

The genetic background is considered in only 5–10% of the cases. The majority of PD is genetically complex, caused by the combination of common genetic variants in concert with environmental factors. Genome-wide association studies have identified twenty-six PD risk loci to date; however, these show only moderate effects on the risk for PD [11]. Furthermore, an individual’s risk of PD is partially the product of as-yet poorly defined polygenic risk factors. Based on the progression of both symptoms and pathology from the periphery to the CNS, Braak and colleagues proposed that, in some patients, Parkinson’s disease might initiate at peripheral sites, particularly the nasal cavity and/or the intestine [12,13]. Very recent epidemiological evidence suggests that both vagotomy and appendectomy decrease the risk of PD in humans. These exciting observations provide further support that peripheral signals, particularly from the gastrointestinal tract, can contribute to the manifestation of this CNS disease [14,15,16]. A large number of studies have been performed to catalog the diversity and abundance of microbial taxa present in the intestinal microbiomes of persons with PD in comparison to healthy individuals [17]. Among these studies, the majority utilized fecal sampling and 16S rRNA sequencing to assess the gastrointestinal microbiome architecture. Most consistently, decreases in the Lachnospiraceae family, including *Blautia* sp. and *Roseburia* sp., as well as decreases in Faecalibacterium, have been reported. Taxa enriched in PD include the Lactobacillaceae family, *Akkermansia* sp., and *Bifidiobacterium* sp. Autopsy studies, including both Parkinson’s disease patients and matched controls, demonstrated that α-synuclein aggregates in Parkinson’s disease patients can be found in both the substantia nigra and the enteric nervous system. Therefore, it has been hypothesized that the pathological process that leads eventually to Parkinson’s disease might initially take place in the enteric nervous system years before the appearance of motor features. Dysbiosis of the normal gut microbiome is thought to be associated with pathophysiologic changes not only in the gastrointestinal system itself but also in the enteric and central nervous systems. [18,19]. Growing evidence indicates that gut microbiota play a critical role in regulating the progression of neurodegenerative diseases such as Parkinson’s disease.

## 2. Hypothesis

We hypothesized that exposure to different classes of antibiotics is involved in the pathogenesis of PD as a contributor to the development of PD-related dysbiosis. Based on the data listed above, it is likely that certain antibiotics change the gut microbiome, favoring curli-producing species. These bacteria deposit αSyn in the enteric nervous system (ENS) and promote further amyloid deposition via cross-seeding, which results in the formation of transmissible self-propagating prion-like proteins. Amyloidosis appears in the ENS and later on in the central nervous system until the full expression of motor symptoms of PD develops, due to the loss of dopamine supply in basal ganglia. In addition, antibiotics result in low-grade systemic inflammation, which also contributes to damage of neurons in the enteric and central nervous systems. As Parkinson’s disease (PD) is a neurodegenerative amyloid disorder with debilitating motor symptoms due to the loss of dopaminesynthesizing, basal-ganglia-projecting neurons in the substantia nigra, an interesting feature of the disease is that most PD patients have gastrointestinal problems and bacterial dysbiosis years before the full expression of motor symptoms [18,19]. We hypothesized that antibiotic consumption might be a contributing factor to gut microbiome dysbiosis in PD, favoring curli-producing Enterobacteria. Curli is a bacterial α-synuclein (αSyn) that is deposited first in the enteric nervous system, and amyloid deposits are propagated in a prion-like manner to the central nervous system. In addition, antibiotics result in low-grade systemic inflammation, which also contributes to damage of neurons in the enteric and central nervous system.

## 3. Materials and Methods

To evaluate the hypothesis, antibiotic consumption databases were compared to PD prevalence data for 2016. For antibiotic consumption patterns, publicly available antibiotic databases (ECDC yearly reports) for 1997–2016 (https://www.ecdc.europa.eu/en/antimicrobial-consumption/database/quality-indicators, accessed on 15 July 2022) were compared to the prevalence and change of the prevalence between 1990–2016 and the Parkinson’s-related death rates were calculated from the Parkinson’s database [1] for 30 European countries. Average yearly consumption of total systemic antibiotics (ATC classification J01) expressed in Defined Daily Dose/1000 Inhabitants/Day (DID) was calculated, as was the relative share of major antibiotic classes consumed in the community at ATC levels 3 and 4 as tetracycline (J01A), penicillin (J01C), broad-spectrum, beta-lactamase sensitive penicillin (J01CA), narrow spectrum, beta-lactamase sensitive penicillin (J01CE), narrow spectrum, beta-lactamase resistant penicillin (J01CF), broad-spectrum, beta-lactamase resistant combination penicillin (J01CR), cephalosporin (J01D), macrolide and lincosamides, streptogramins (J01F), and quinolone (J01M). Estimating the possible time-lapse of the antibiotic consumption and the development of PD, we separately calculated the average antibiotic consumption for the periods of 1997–2001, 2002–2006, 2007–2011, and 2012–2016, covering 20 years. Results were featured in Table 1, Table 2, Table 3 and Table 4. A separate comparison was performed between the antibiotic consumption patterns of countries with a declining prevalence of PD (Bulgaria, France, Italy, and the Netherlands) and the countries with the highest prevalence of PD (Denmark, Norway, Portugal, and the UK), featured in Table 5. Diagrams were plotted for graphically demonstrating the association between the increases of PD prevalence in 21 European countries and the average consumption of narrow-spectrum penicillin for the time segment 1997–2001. (Figure 1 and Figure 2).

Pearson calculation was applied to estimate the correlation between antibiotic consumption and the prevalence, change of the prevalence (1990–2016), and death prevalence attributed to PD. A significant correlation was considered when *p* values were ≤0.05. A non-significant correlation was estimated when the *p* values fell between 0.051–0.09. Positive (supportive) and negative (inhibitor) significant correlations were considered and evaluated. Statistical results were recorded and featured in the same tables (Table 1, Table 2, Table 3 and Table 4). Logistic regression analysis was performed to estimate the odds ratio (OR) at the 95% confidence interval (CI) together with the related *p*-values.

## 4. Results

The results are summarized in Table 1, Table 2, Table 3, Table 4 and Table 5. Across different time sections, the possible promoting effect of the narrow-spectrum penicillin (J01CE, J01CF) on the development of PD was well observed, while the inhibitory effect of broad-spectrum penicillin (J01CA), broad-spectrum combination penicillin (J01CR), and other broad-spectrum antibiotics (cephalosporin/J01D/, quinolone/J01M/) were detected also. In the first period (Table 1 1997–2001), a significant promoting effect was observed between the prevalence of PD (2016) and the consumption of narrow-spectrum penicillin (J01CE, J01CF) and cephalosporin (J01D). A significant inhibitory effect was observed between broad-spectrum, beta-lactamase sensitive penicillin (J01CA), quinolone (J01M), and the prevalence increase of PD (1990–2016). It is of concern that quinolone consumption showed positive significant association with the PD-attributed death rate. The other periods (Table 2, Table 3 and Table 4, 2002–2006, 2007–2011, 2012–2016) showed similar associations. It was observed that countries with a reduction in PD prevalence between 1990–2016 (Bulgaria, France, Italy, Netherlands) considerably reduced their consumption of narrow-spectrum penicillin (J01CE, J01CF) in this period. In comparison, the similar four countries with the highest increase in PD prevalence (Denmark, Norway, Portugal, UK), it was observed that they used a nearly six-fold higher amount of narrow-spectrum, beta-lactamase sensitive (J01CE) and three-fold higher of beta-lactamase-resistant penicillin (J01CF) (Table 5). Diagrams (Figure 1) indicate a strong correlation between the average consumption of narrow-spectrum penicillin (J01CE + J01CF) and the prevalence increase of PD (N: 21). Figure 2 indicates the association between the rank order of PD prevalence increase and the consumption of narrow-spectrum antibiotics.

## 5. Discussion

Our observations indicate a positive statistical association and higher risk for the development of PD in countries consuming higher amounts of narrow-spectrum penicillin, which is demonstrated in Figure 2, showing the association between the PD prevalence increase and the consumption of narrow-spectrum penicillin for the time period 1997–2001. Similar associations were detected in the other time periods as well, but statistically significant negative associations and reduced risk of PD were also detected on comparing the consumption of broad-spectrum antibiotics (quinolone) and the PD prevalence. The opposing effect of the consumption of “promoter” (penicillin) and “inhibitor” antibiotics (broad-spectrum antibiotics) might determine the actual prevalence of PD, along with other factors playing a role in the development of PD (genetics).

While Parkinson’s disease has been historically studied as a disease of the central nervous system, there is a growing appreciation for the roles of both gastrointestinal function and its resident microbes within this disease state. Recent studies focused on the microbiome during Parkinson’s disease may advance our understanding of disease etiology and provide perspective for previously unrecognized therapeutic avenues through the modulation of intestinal microbes. In recent years, a large number of studies have been performed to catalog the diversity and abundance of microbial taxa present in the intestinal microbiomes of persons with PD in comparison to healthy individuals [20]. Among these studies, the majority utilized fecal sampling and 16S rRNA sequencing to assess the gastrointestinal microbiome architecture. Most consistently, decreases in the Lachnospiraceae family, including *Blautia* sp. and *Roseburia* sp., as well as decreases in *Faecalibacterium*, have been reported. Taxa enriched in PD include the *Lactobacillaceae* family, *Akkermansia* sp., and *Bifidiobacterium* sp. Fecal-derived microbial communities are easily observed. Unlike tissue or small intestinal contents, they do not require invasive procedures to acquire. However, the microbiome present in the feces is not identical to that within other regions of the gastrointestinal tract. It is critical to determine whether the dysbiosis present during PD has functional contributions, with microbes as active instigators or modulators of the disease state. Alternatively, dysbiosis may arise due to inherent changes in the host’s physiology (e.g., functional constipation) and the microbial population is affected as a bystander. In the latter case, dysbiosis would not be causative or influential on PD, but instead be an epiphenomenon. Experimental evidence supporting a functional role for gut microbes in impacting disease outcomes in PD is beginning to surface. In one study, eliminating microbial signals in ⍺Syn-overexpressing mice through either germ-free rederivation or microbial depletion with antibiotics decreased ⍺Syn amyloid deposition in the brain, neuroinflammation, and motor deficits. Antibiotic treatment has been shown to also dampen MPTP (1-methyl-4-phenyl-1,2,3,6-tetrahydropyridine) in toxin-induced models of PD by preventing ⍺Syn aggregation and dopaminergic cell death [21,22,23]. Similarly, experimental studies demonstrated the inhibitory effect of Doxycyclin in the development of α-synuclein-induced pathogenicity, which is in accord with our observation (Table 4) that tetracyclines might have an inhibitory effect on the development of Parkinson’s disease [24]. Similar microbial modulation of pathology has been observed in other models of amyloid disease. In two different mouse models of amyloid-β deposition, representing Alzheimer’s disease, the germ-free status of antibiotic treatment decreased amyloid-β pathology in relevant brain regions. It is exciting to consider the possibility of shared mechanisms of microbial influence within the gastrointestinal tract on the deposition of amyloid proteins in the brain that is central to neurodegenerative disease [25,26,27]. Analysis of the microbial community composition revealed that those animals displaying the most motor dysfunction (i.e., those with ⍺Syn-overexpression and receiving PD-derived microbes) harbored several changes to microbial taxa. These alterations may suggest the presence of microbes that exacerbate disease in the model, or instead, the absence of microbes that may dampen a disease state. It is worthy of consideration that *E. coli* produces an extracellular structure, termed curli, which takes on an amyloid form similar to pathogenic ⍺Syn [28]. Curli amyloids provide a mechanism of surface attachment, immune evasion, and defense against bacteriophages. This suggests that, rather than LPS signaling alone, the production of this bacterial amyloid could provide a specific signal from the GI tract that can impact PD-relevant physiologies in the brain. How curli may act to elicit these changes is largely unknown. Curli may act to stimulate inflammatory responses or, based on its structure, may have the potential to template the generation of ⍺Syn amyloids and directly drive PD pathology [29]. There also exists the possibility of beneficial interactions that are missing or decreased due to PD dysbiosis [30]. There may be microbes that produce molecules that dampen inflammation or inhibit the aggregation of ⍺Syn, ultimately preventing disease initiation or slowing its progression. There are many examples of beneficial microbes that are capable of specifically dampening inflammatory responses within the GI tract [31,32], as we have observed in the inhibitory effect of certain classes of antibiotics (Table 1, Table 2, Table 3 and Table 4, green filling color). In line with this, one experimental study of fecal microbial transplant (FMT) demonstrated that mice receiving a healthy microbial population were protected from dopaminergic neuron death following MPTP treatment [33]. While data demonstrating that persons with PD harbor a dysbiotic intestinal microbiome continue to accumulate, one pressing question is how the dysbiosis initially arises. Certainly, physiological changes such as constipation, which occurs early in PD, can shift the population of the gut microbiome [34,35]. Diet is also well established to shape the intestinal microbiome; however, links between diet and PD are tenuous [36]. Pharmaceutical interventions also affect the microbiome structure, but even with this confounding variable taken into account, dysbiosis during PD is still apparent [37]. The microbiome of persons with PD is indeed enriched for microbial genes that are involved in the metabolism of xenobiotics, suggesting that environmental exposures impact microbiome structure [38].

## 6. Conclusions

Our comparative analysis of antibiotic consumption patterns for the years 1997–2016 in 30 European countries and the PD database established for 2016 indicated a potential sequential role of different classes of antibiotics in the development of PD, possibly through the modification of gut flora, either augmenting (narrow-spectrum penicillin) or inhibiting (broad-spectrum antibiotics) the process. The augmenting effect of the narrow-spectrum penicillin was observed in every time segment of the compared databases of antibiotic consumption throughout the 20 years (1997–2016), but the effect of other antibiotic classes showing either augmenting or inhibiting properties occurred only in different time-segments. Countries with a high consumption rate of narrow-spectrum penicillin, showed a high prevalence increase of PD, and countries experiencing a reduction in PD prevalence between 1990–2016 showed three–six-fold lower consumption of narrow-spectrum penicillin. It could be hypothesized that penicillin augments the production of curli amyloid fibril or promotes the proliferation of curli-producing microbes and other antibiotic classes might act through the upregulation or down-regulation of the curli-producing capabilities of the gut flora influencing the pathological process, which leads to the development of PD. It might be suspected also that the altered microbiome produces less SCFA (short-chained fatty acids), as SCFA induces regulatory T (Treg) cells; a decrease in SCFA-producing bacteria may be a prerequisite for the development of PD. Similar observations had been reported in the literature [39,40,41]. Our observation indicated the possible role of antibiotic consumption in the development of PD and our results are further strengthened by the fact that countries with high consumption of narrow-spectrum penicillin experienced a higher prevalence increase of PD, while when others reducing or ceasing the consumption of narrow-spectrum penicillin experienced a reduction in the prevalence of PD in the previous 25 years. Further studies might elucidate the molecular background of this mechanism.

**Limitations:** The associations of PD’s prevalence with different classes of antibiotics could not be applied to individual cases.

**Strength:** Using large databases, our statistical results are convincing and in accord with other publications.

## Figures and Tables

**Figure 1 antibiotics-11-01145-f001:**
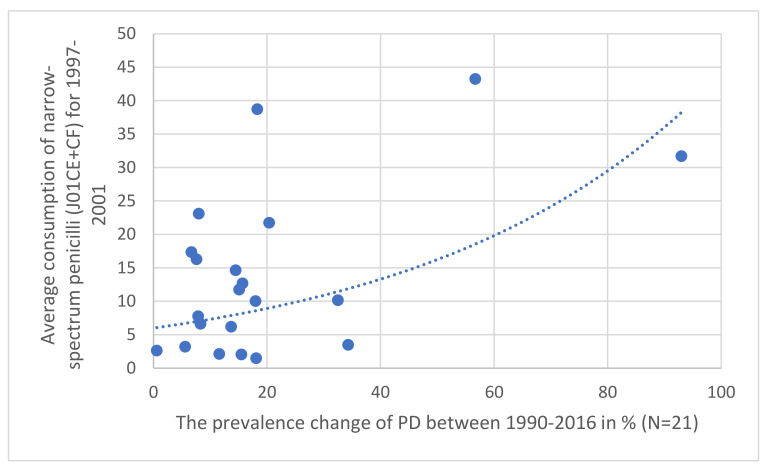
Correlation observed between the prevalence increase of PD in 21 European countries (1997–2001) compared to the relative share of the consumption of narrow-spectrum, beta-lactamase sensitive (J01CE) and resistant (J01CF) penicillin (J01CE + CF).

**Figure 2 antibiotics-11-01145-f002:**
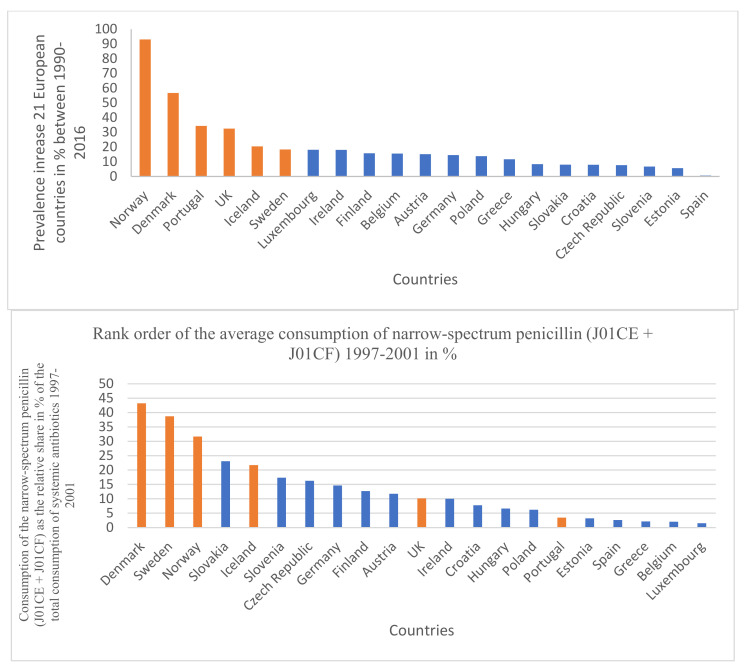
Rank order of prevalence increase of PD in 21 European countries (%) for the time period 1997–2011 and the rank order of the consumption of narrow-spectrum penicillin (J01CE + J01CF). Identical countries are marked with orange filling color. Out of six countries with the highest increase of PD prevalence, four countries are identical with the penicillin-consumption rank order (1997–2001).

**Table 1 antibiotics-11-01145-t001:** Average antibiotic consumption for 1997–2001 expressed as a relative share in % of the total systemic antibiotic consumption in the community estimated in Defined Daily Dose/1000 Inhabitants/Day (DID) compared to the prevalence of PD estimated for 2016, the change of prevalence between 1990–2016 in percentage (%), and the PD-related deaths/100,000 population. Positive, significant correlation or significantly elevated risk was estimated when the *p*-value showed ≤0.05 (*p* = ≤0.05) (marked with yellow filling color) and the negative correlation or lowered risk was marked with green filling color. Positive/negative, non-significant correlations or elevated/lowered risks were considered when the *p*-value fell between 0.051 and 0.09 (*p* = 0.051 ≤ 0.09) and marked with an orange filling color.

Antib. Cons. 1997–2001 in %	PD/100,000	PD-Related Deaths/100,000 Population	The Prevalence Change of PD between 1990–2016 in %	J01 in DID	J01A%	J01C%	J01CA%	J01CE%	J01CF%	J01CR%	J01D%	J01F%	J01M%
Austria	181.137	8.481	15.1	11.525	11.171	29.718	7.462	11.649	0.087	10.629	13.427	29.154	10.390
Belgium	183.551	8.578	15.5	22.275	16.173	29.630	11.582	0.774	1.257	15.881	17.587	16.263	9.944
Bulgaria	238.176	9.364	–5.3	19.333	19.207	47.069	28.362	15.431	0.397	2.862	11.052	2.879	4.948
Croatia	232.584	9.677	7.9	16.500	8.848	40.606	17.030	7.333	0.424	15.576	19.697	11.697	8.303
Cyprus	14.565	0.679	4.9										
Czech Republic	214.117	8.186	7.6	16.800	17.917	38.690	14.494	15.506	0.774	7.679	7.768	14.673	6.101
Denmark	157.737	7.149	56.7	12.125	8.000	59.588	16.371	38.990	4.247	0.165	0.206	17.732	1.567
Estonia	23.396	0.920	5.6	14.400	20.069	38.194	31.319	2.847	0.347	3.611	4.722	10.069	5.694
Finland	186.397	9.540	15.7	17.975	25.438	25.730	11.558	12.350	0.320	1.530	12.420	11.127	3.936
France	179.718	8.651	–5.1	28.400	10.968	41.373	27.790	0.819	1.910	10.863	15.995	20.493	7.245
Germany	195.949	8.824	14.5	12.425	23.501	29.577	13.803	14.467	0.161	0.986	7.586	19.698	7.887
Greece	212.293	9.910	11.6	25.125	10.965	26.567	14.886	2.080	0.030	9.602	25.433	25.234	6.905
Hungary	213.399	8.390	8.3	17.675	14.455	37.907	15.021	6.620	0.000	16.280	15.318	19.066	5.926
Iceland	140.112	6.503	20.4	21.325	23.447	49.472	20.059	15.498	6.225	7.691	2.626	7.913	2.943
Ireland	125.681	5.257	18	15.425	20.340	42.139	18.639	5.154	4.862	13.468	12.334	13.841	3.712
Italy	238.666	10.761	–5.5	21.500	2.450	35.659	22.000	0.233	0.124	13.349	17.349	23.860	12.729
Latvia	236.552	9.128	7.7										
Lithuania	237.895	9.340	9.2										
Luxembourg	147.791	6.772	18.1	24.700	12.571	28.543	14.140	0.668	0.820	12.986	21.964	20.435	8.381
Malta	163.488	6.585	15.3										
Netherlands	194.930	8.588	–6.0	8.875	27.887	31.549	14.789	5.803	2.648	8.225	1.014	13.662	9.606
Norway	142.955	6.504	93	15.150	20.759	41.914	10.363	30.198	1.485	0.066	1.584	10.858	1.980
Poland	197.259	7.750	13.7	20.000	20.025	35.250	23.475	5.813	0.375	5.563	10.713	11.325	5.463
Portugal	179.406	8.167	34.3	20.950	7.804	37.589	15.239	0.227	3.246	19.057	15.764	17.041	15.418
Romania	206.316	8.173	8.1										
Slovakia	175.207	6.587	8	24.300	8.546	53.498	19.451	22.867	0.219	11.029	9.849	13.635	6.200
Slovenia	231.957	9.681	6.7	15.950	5.157	51.097	13.088	16.552	0.784	20.862	3.918	22.743	9.420
Spain	199.826	9.378	0.6	16.075	4.121	44.012	21.742	0.964	1.649	19.627	14.852	19.922	13.515
Sweden	197.855	9.214	18.3	15.325	21.582	46.166	6.509	31.680	7.031	1.011	3.670	6.346	6.933
UK	176.035	8.263	32.5	13.275	24.463	37.665	21.864	5.009	5.141	5.687	6.234	18.512	3.315
PD (Pearson) R	**Antibiotic consumption 1997–2001**			0.113	−0.304	−0.048	−0.257	−0.011	−0.240	0.279	0.300	0.207	0.340
PD (Pearson) *p*				0.592	0.139	0.820	0.214	0.960	0.248	0.176	0.145	0.322	0.096
PD OR				1.052	0.920	0.970	0.992	0.962	**0.627**	*1.111*	**1.138**	1.073	*1.223*
PD CI95%				0.907–1.229	0.823–1.019	0.887–1.059	0.880–1.119	0.892–1.033	**0.392–0.924**	*0.986–1.266*	**1.007–1.307**	0.947–1.232	*0.984–1.563*
PD p				0.508	0.120	0.497	0.896	0.281	**0.028**	*0.093*	**0.047**	0.284	*0.081*
PD% (Pearson) R	**Prevalence change of PD 1990–2016 in %**			−0.224	0.135	0.195	**−0.373**	**0.546**	0.303	*−0.375*	*−0.361*	−0.100	**−0.433**
PD% (Pearson) *p*				0.281	0.521	0.350	**0.067**	**0.005**	0.140	*0.065*	*0.077*	0.634	**0.031**
PD% OR				0.945	1.073	0.998	**0.882**	1.061	**1.626**	0.923	0.947	0.995	*0.789*
PD% CI95%				0.802–1.110	0.965–1.199	0.917–1.085	**0.774–0.993**	0.988–1.146	**1.111–2.585**	0.820–1.033	0.843–1.057	0.881–1.119	*0.607–1.007*
PD% *p*				0.488	0.199	0.967	**0.044**	0.107	**0.021**	0.168	0.340	0.933	*0.064*
PD death (Pearson) R	**Death rate attributed to PD/** **100,000 population**			0.390	0.306	0.342	0.239	−0.143	−0.057	*−0.346*	−0.175	−0.281	0.121
PD death Pearson) *p*				0.054	0.137	0.095	0.249	0.495	0.787	*0.090*	0.402	0.174	0.565
PD death OR				0.999	1.070	1.047	1.049	1.035	1.214	0.928	*0.902*	0.950	**0.770**
PD death CI95%				0.866–1.155	0.965–1.195	0.956–1.151	0.929–1.191	0.960–1.122	0.835–1.833	0.818–1.045	*0.794–1.013*	0.828–1.078	**0.590–0.965**
PD death *p*				0.986	0.207	0.324	0.441	0.365	0.324	0.226	*0.091*	0.441	**0.034**

**Table 2 antibiotics-11-01145-t002:** Average antibiotic consumption for 2002–2006 expressed as a relative share in % of the total systemic antibiotic consumption in the community estimated in Defined Daily Dose/1000 Inhabitants/Day (DID) compared to the prevalence of PD estimated for 2016, the change of prevalence between 1990–2016 in percentage (%), and the PD-related deaths/100,000 population. Positive, significant correlation or significantly elevated risk was estimated, when the *p*-value showed ≤0.05 (*p* = ≤0.05) (marked with yellow filling color) and the negative correlation or lowered risk was marked with green filling color. Positive/negative, non-significant correlations or elevated/lowered risks were considered when the *p*-value fell between 0.051 and 0.09 (*p* = 0.051 ≤ 0.09) and marked with an orange filling color.

Antib. Cons. 2002–2006 in %	PD/100,000	PD-Related Deaths/100,000 Population	The Prevalence Change of PD between 1990–2016 in %	J01 in DID	J01A%	J01C%	J01CA%	J01CE%	J01CF%	J01CR%	J01D%	J01F%	J01M%
Austria	181.137	8.481	15.1	11.700	9.966	34.017	6.991	9.094	0.085	17.726	14.598	26.735	15.419
Belgium	183.551	8.578	15.5	20.250	11.062	35.432	12.704	0.741	1.160	20.914	17.790	13.642	12.802
Bulgaria	238.176	9.364	–5.3	16.020	17.266	43.446	29.126	6.841	0.125	7.303	9.413	7.990	8.252
Croatia	232.584	9.677	7.9	20.240	9.506	42.292	16.443	8.034	0.316	17.599	18.182	11.996	9.051
Cyprus	14.565	0.679	4.9	27.100	11.476	35.793	16.458	0.443	0.111	18.893	23.063	12.177	14.133
Czech Rep.	214.117	8.186	7.6	8.720	25.849	48.394	14.266	18.463	0.436	15.229	10.940	26.330	15.849
Denmark	157.737	7.149	56.7	13.700	8.642	61.460	16.701	37.825	6.642	0.336	0.219	16.263	4.934
Estonia	23.396	0.920	5.6	9.940	24.909	31.187	23.360	3.099	0.000	4.829	6.459	14.306	8.833
Finland	186.397	9.540	15.7	16.840	22.648	25.772	12.542	9.561	0.273	3.385	11.330	11.734	6.295
France	179.718	8.651	–5.1	24.440	13.871	40.507	21.989	0.745	1.899	15.917	13.822	18.674	10.540
Germany	195.949	8.824	14.5	12.580	25.469	26.232	15.056	9.984	0.143	1.224	7.774	18.251	11.399
Greece	212.293	9.910	11.6	30.660	8.500	24.984	11.892	2.153	0.020	10.946	20.522	30.972	11.944
Hungary	213.399	8.390	8.3	15.980	11.039	37.797	12.190	6.270	0.000	19.249	15.181	20.438	12.791
Iceland	140.112	6.503	20.4	19.480	26.016	48.563	17.310	14.251	6.561	10.462	3.932	7.649	4.630
Ireland	125.681	5.257	18	17.460	20.057	41.695	14.124	4.559	5.063	17.766	12.577	16.861	7.056
Italy	238.666	10.761	–5.5	21.640	2.089	39.649	20.712	0.065	0.092	18.771	15.832	22.847	15.980
Latvia	236.552	9.128	7.7	10.125	24.025	37.284	27.136	1.679	0.049	8.543	5.605	9.358	9.951
Lithuania	237.895	9.340	9.2	22.100	7.783	65.611	26.787	31.946	3.529	3.348	3.937	4.977	3.756
Luxembourg	147.791	6.772	18.1	22.820	10.070	33.742	12.463	0.684	0.806	19.825	21.341	15.837	11.805
Malta	163.488	6.585	15.3										
Netherlands	194.930	8.588	–6.0	9.060	25.541	32.009	13.223	4.967	3.113	10.596	2.715	15.475	10.751
Norway	142.955	6.504	93	15.400	19.078	41.039	11.909	26.052	2.896	0.026	1.299	12.429	4.766
Poland	197.259	7.750	13.7	13.100	18.435	32.443	25.878	2.042	0.095	4.351	9.542	17.156	10.897
Portugal	179.406	8.167	34.3	20.800	5.231	39.327	10.644	0.221	3.115	25.394	17.712	18.654	16.510
Romania	206.316	8.173	8.1										
Slovakia	175.207	6.587	8	22.100	10.380	47.240	13.638	20.362	0.154	13.158	11.783	17.692	9.213
Slovenia	231.957	9.681	6.7	13.840	4.538	51.879	14.899	16.329	1.156	19.436	7.760	21.720	11.503
Spain	199.826	9.378	0.6	15.200	4.000	48.553	17.934	0.605	1.447	28.553	16.474	16.421	14.882
Sweden	197.855	9.214	18.3	14.580	21.619	44.307	7.037	28.080	8.285	0.988	2.510	5.967	6.680
UK	176.035	8.263	32.5	13.360	24.880	38.772	22.964	4.955	6.677	4.985	5.674	17.021	6.302
PD (Pearson) R	**Antibiotic consumption 2002–2006**			−0.101	−0.218	0.203	0.105	0.120	−0.093	0.034	−0.092	0.123	0.085
PD (Pearson) *p*				0.610	0.265	0.300	0.596	0.543	0.638	0.865	0.643	0.532	0.667
PD OR				0.999	0.941	1.021	1.075	0.988	**0.735**	1.031	1.042	1.080	1.166
PD CI95%				0.874–1.141	0.856–1.030	0.945–1.107	0.957–1.215	0.921–1.061	**0.527–0.983**	0.949–1.124	0.928–1.174	0.958–1.232	0.960–1.437
PD *p*				0.991	0.193	0.595	0.229	0.737	**0.048**	0.472	0.491	0.221	0.131
PD% (Pearson)R	**Prevalence change of PD 1990–2016 in %**			−0.092	0.078	0.152	−0.285	**0.491**	**0.431**	**−0.386**	*−0.370*	−0.097	**−0.415**
PD% (Pearson) *p*				0.642	0.692	0.441	0.142	**0.008**	**0.022**	**0.042**	*0.052*	0.623	**0.028**
PD% OR				1.016	1.007	**0.921**	*0.886*	0.978	0.764	1.004	*1.108*	1.046	1.075
PD% CI95%				0.897–1.150	0.922–1.100	**0.842–0.993**	*0.771–1.003*	0.913–1.045	0.522–1.053	0.921–1.097	*0.991–1.251*	0.934–1.175	0.891–1.309
PD% *p*				0.797	0.875	**0.044**	*0.067*	0.511	0.123	0.921	*0.080*	0.437	0.453
PD death (Pears.) R	**Death rate attributed to PD/** **100,000 population**			−0.034	−0.238	0.099	−0.024	0.072	−0.012	0.045	−0.049	0.161	0.090
PD death (Pears.) *p*				0.864	0.223	0.615	0.902	0.717	0.950	0.820	0.804	0.413	0.647
PD death OR				0.991	1.058	0.989	0.960	1.011	1.138	0.979	0.978	1.007	0.936
PD death CI95%				0.873–1.125	0.969–1.160	0.912–1.069	0.848–1.083	0.940–1.088	0.852–1.557	0.897–1.068	0.872–1.095	0.896–1.134	0.771–1.127
PD death *p*				0.883	0.2130	0.7740	0.5090	0.7590	0.3910	0.6320	0.7010	0.8990	0.4840

**Table 3 antibiotics-11-01145-t003:** Average antibiotic consumption for 2007–2011 expressed as a relative share in % of the total systemic antibiotic consumption in the community estimated in Defined Daily Dose/1000 Inhabitants/Day (DID) compared to the prevalence of PD estimated for 2016, the change of prevalence between 1990–2016 in percentage (%), and the PD-related deaths/100,000 population. Positive, significant correlation or significantly elevated risk was estimated, when the *p*-value showed ≤0.05 (*p* = ≤0.05) (marked with yellow filling color) and the negative correlation or lowered risk was marked with green filling color. Positive/negative, non-significant correlations or elevated/lowered risks were considered when the *p*-value fell between 0.051 and 0.09 (*p* = 0.051 ≤ 0.09) and marked with an orange filling color.

Antib. Cons 2007–2011 in %	PD/100,000	PD-Related Deaths/100,000 Population	The Prevalence Change of PD between 1990–2016 in %	J01 in DID	J01A%	J01C%	J01CA%	J01CE%	J01CF%	J01CR%	J01D%	J01F%	J01M
Austria	181.137	8.481	15.1	13.140	9.482	35.921	6.743	7.336	0.076	21.659	13.029	27.641	10.228
Belgium	183.551	8.578	15.5	22.520	9.316	46.092	19.583	0.400	1.110	24.973	8.268	12.789	11.297
Bulgaria	238.176	9.364	–5.3	16.580	11.665	36.671	24.029	2.400	0.000	10.205	13.522	18.782	12.364
Croatia	232.584	9.677	7.9	18.880	8.369	38.771	12.839	5.890	0.117	19.905	19.121	17.193	7.394
Cyprus	14.565	0.679	4.9	27.960	10.179	36.981	13.720	0.393	0.093	22.775	22.654	12.310	14.385
Czech Rep.	214.117	8.186	7.6	15.980	15.094	36.295	7.472	12.804	0.188	15.657	8.861	21.665	7.447
Denmark	157.737	7.149	56.7	15.620	10.346	61.460	18.592	33.496	7.362	2.061	0.218	15.544	3.278
Estonia	23.396	0.920	5.6	10.280	20.564	30.739	19.630	2.626	0.019	8.599	8.482	22.101	8.191
Finland	186.397	9.540	15.7	17.160	24.231	28.438	15.152	8.357	0.093	4.977	13.671	9.196	5.152
France	179.718	8.651	–5.1	23.620	13.887	44.793	24.234	0.627	1.719	18.196	11.550	16.926	8.527
Germany	195.949	8.824	14.5	13.420	22.325	23.398	15.231	6.408	0.075	1.624	16.751	17.794	10.879
Greece	212.293	9.910	11.6	36.220	6.444	24.903	12.071	2.457	0.006	10.370	24.197	29.680	7.714
Hungary	213.399	8.390	8.3	13.620	10.000	34.949	8.722	4.743	0.000	21.424	13.671	22.100	13.172
Iceland	140.112	6.503	20.4	18.460	27.541	47.562	15.255	12.979	6.132	13.131	2.048	8.277	4.442
Ireland	125.681	5.257	18	18.680	15.664	43.790	12.612	4.882	5.546	20.824	7.784	21.231	5.203
Italy	238.666	10.761	–5.5	23.260	2.244	43.766	16.028	0.000	0.043	27.687	11.522	22.029	15.116
Latvia	236.552	9.128	7.7	10.040	24.183	37.649	26.056	0.976	0.000	10.677	5.219	11.375	9.741
Lithuania	237.895	9.340	9.2	17.360	11.717	44.355	28.687	8.594	0.081	7.016	9.988	10.806	7.488
Luxembourg	147.791	6.772	18.1	23.280	8.832	37.973	13.024	0.412	0.765	23.789	17.938	16.435	12.079
Malta	163.488	6.585	15.3	17.940	5.819	35.117	4.426	0.279	0.323	29.989	26.856	19.643	9.777
Netherlands	194.930	8.588	–6.0	10.020	26.208	31.737	12.475	3.952	3.653	11.776	0.419	14.870	8.842
Norway	142.955	6.504	93	15.360	18.529	41.536	13.242	24.401	3.815	0.000	0.872	12.357	3.359
Poland	197.259	7.750	13.7	18.480	12.900	38.312	19.394	0.790	0.032	18.041	13.182	21.255	6.569
Portugal	179.406	8.167	34.3	15.180	4.308	43.083	8.590	0.145	2.806	31.647	9.763	19.104	15.520
Romania	206.316	8.173	8.1	18.050	4.017	47.091	20.028	3.740	3.102	20.166	18.089	13.075	12.853
Slovakia	175.207	6.587	8	21.375	6.994	33.801	7.778	8.772	0.000	17.158	18.456	28.140	9.743
Slovenia	231.957	9.681	6.7	12.320	2.370	57.792	18.360	16.153	1.055	22.305	3.458	18.571	8.929
Spain	199.826	9.378	0.6	16.040	3.978	52.494	18.691	0.561	1.209	32.045	10.100	12.369	15.499
Sweden	197.855	9.214	18.3	14.140	23.324	47.666	8.076	28.614	9.859	1.160	1.782	4.894	5.771
UK	176.035	8.263	32.5	15.520	25.322	39.304	22.410	4.742	7.423	4.807	3.943	16.675	3.247
PD (Pearson) R	**Antibiotic consumption 2007–2011**			−0.108	−0.195	0.104	0.152	0.016	−0.130	0.018	−0.056	0.039	0.082
PD (Pearson) *p*				0.569	0.303	0.585	0.424	0.934	0.493	0.924	0.770	0.838	0.667
PD OR				1.000	0.938	0.986	1.075	0.960	0.774	1.011	1.036	1.053	1.157
PD CI95%				0.871–1.139	0.857–1.021	0.910–1.068	0.964–1.207	0.882–1.041	0.578–0.993	0.943–1.085	0.937–1.151	0.943–1.182	0.960–1.414
PD *p*				0.997	0.148	0.730	0.199	0.319	0.058	0.754	0.499	0.365	0.134
PD% (Pearson) R	**Prevalence change of PD 1990–2016 in %**			−0.114	0.157	0.225	−0.141	**0.607**	**0.461**	**−0.393**	**−0.362**	−0.180	**−0.501**
PD% (Pearson) *p*				0.548	0.408	0.231	0.458	**< .001**	**0.010**	**0.032**	**0.049**	0.340	**0.005**
PD% OR				1.005	1.061	1.021	0.926	**1.099**	**1.489**	0.951	0.952	0.965	**0.734**
PD% CI95%				0.891–1.125	0.969–1.165	0.945–1.106	0.828–1.030	**1.012–1.208**	**1.128–2.095**	0.881–1.024	0.862–1.047	0.866–1.073	**0.582–0.906**
PD% *p*				0.935	0.202	0.605	0.162	**0.032**	**0.010**	0.186	0.321	0.510	**0.006**
PD death R	**Death rate attributed to PD/** **100,000 population**			−0.040	−0.141	0.116	0.126	0.045	−0.034	−0.009	−0.103	−0.028	0.045
PD death *p*				0.833	0.457	0.542	0.505	0.813	0.860	0.963	0.588	0.883	0.812
PD death OR				1.016	1.024	0.994	0.918	1.019	1.146	1.009	1.010	1.058	0.897
PD death CI95%				0.849–1.168	0.940–1.118	0.914–1.078	0.818–1.023	0.933–1.117	0.884–1.532	0.939–1.087	0.914–1.117	0.938–1.203	0.738–1.081
PD death *p*				0.814	0.585	0.886	0.129	0.661	0.322	0.806	0.843	0.365	0.257

**Table 4 antibiotics-11-01145-t004:** Average antibiotic consumption for 2012–2016 expressed as a relative share in % of the total systemic antibiotic consumption in the community estimated in Defined Daily Dose/1000 Inhabitants/Day (DID) compared to the prevalence of PD estimated for 2016, the change of prevalence between 1990–2016 in percentage (%), and the PD-related deaths/100,000 population. Positive, significant correlation or significantly elevated risk was estimated, when the *p*-value showed ≤0.05 (*p* = ≤0.05) (marked with yellow filling color) and the negative correlation or lowered risk was marked with green filling color. Positive/negative, non-significant correlations or elevated/lowered risks were considered when the *p*-value fell between 0.051 and 0.09 (*p* = 0.051 ≤ 0.09) and marked with an orange filling color.

Antib. Cons 2012–2016 in %	PD/100,000	PD-Related Deaths/100,000 Population	The Prevalence of PD between 1990–2016 in %	J01 in DID	J01A%	J01C%	J01CA%	J01CE%	J01CF%	J01CR%	J01D%	J01F%	J01M%
Austria	181.137	8.481	15.1	12.400	8.484	38.548	6.016	6.468	0.065	25.935	12.677	25.194	10.613
Belgium	183.551	8.578	15.5	22.840	9.098	45.184	22.119	0.131	1.112	21.778	6.340	15.228	11.322
Bulgaria	238.176	9.364	–5.3	17.680	9.842	30.317	18.552	1.075	0.000	11.493	18.507	20.441	15.158
Croatia	232.584	9.677	7.9	17.760	6.374	44.482	11.779	3.761	0.011	29.077	16.396	16.318	8.390
Cyprus	14.565	0.679	4.9	25.240	13.177	34.469	9.960	0.309	0.079	24.342	19.889	11.181	17.662
Czech Rep.	214.117	8.186	7.6	16.775	12.399	35.320	7.189	10.656	0.238	17.049	11.028	22.370	5.425
Denmark	157.737	7.149	56.7	15.420	11.154	63.165	20.636	28.223	8.716	5.538	0.195	12.412	3.294
Estonia	23.396	0.920	5.6	10.340	15.222	31.141	16.654	1.818	0.000	12.592	10.851	23.327	8.530
Finland	186.397	9.540	15.7	16.460	25.261	29.769	16.355	7.594	0.267	5.456	13.341	7.217	4.885
France	179.718	8.651	–5.1	23.780	13.759	52.397	29.865	0.698	1.001	20.328	9.041	13.802	7.250
Germany	195.949	8.824	14.5	13.500	15.911	25.037	16.593	5.837	0.074	2.504	22.504	18.711	10.030
Greece	212.293	9.910	11.6	29.540	7.705	30.941	14.448	0.223	0.020	15.850	24.516	24.699	8.280
Hungary	213.399	8.390	8.3	13.680	8.567	33.772	6.740	1.827	0.000	25.219	13.874	21.637	16.959
Iceland	140.112	6.503	20.4	18.400	25.011	48.261	15.293	11.467	5.543	15.522	3.130	9.054	5.272
Ireland	125.681	5.257	18	20.140	13.625	49.454	14.866	5.204	7.795	22.036	5.988	21.122	4.340
Italy	238.666	10.761	–5.5	22.520	2.469	46.448	12.087	0.000	0.044	34.378	10.453	20.515	15.133
Latvia	236.552	9.128	7.7	11.020	20.726	38.838	26.969	0.454	0.018	11.434	4.646	15.445	9.492
Lithuania	237.895	9.340	9.2	13.720	10.525	47.376	36.385	1.152	0.000	10.087	8.149	14.621	6.793
Luxembourg	147.791	6.772	18.1	22.200	8.045	40.000	13.964	0.072	0.676	25.243	15.901	17.225	11.604
Malta	163.488	6.585	15.3	19.580	6.537	31.256	2.411	0.388	0.276	28.151	23.534	19.755	13.279
Netherlands	194.930	8.588	–6.0	9.560	23.849	32.845	14.121	2.950	4.498	11.297	0.397	14.644	8.117
Norway	142.955	6.504	93	15.620	20.205	39.181	14.174	21.140	3.969	0.038	0.589	9.757	3.163
Poland	197.259	7.750	13.7	20.760	11.590	31.310	17.148	1.137	0.039	13.131	12.225	19.268	6.166
Portugal	179.406	8.167	34.3	17.240	4.988	47.332	9.118	0.081	2.633	35.302	8.724	17.274	12.459
Romania	206.316	8.173	8.1	26.340	4.313	47.836	20.273	3.219	2.787	22.521	18.762	10.873	12.863
Slovakia	175.207	6.587	8	20.240	7.925	30.534	5.632	6.364	0.000	18.656	21.443	27.668	9.308
Slovenia	231.957	9.681	6.7	11.580	3.748	60.449	19.603	14.784	1.399	24.905	2.625	15.147	9.655
Spain	199.826	9.378	0.6	18.420	4.810	55.266	18.274	0.423	1.097	32.671	9.262	12.269	13.442
Sweden	197.855	9.214	18.3	12.480	22.260	50.160	8.526	27.244	12.340	1.923	1.234	4.808	5.593
UK	176.035	8.263	32.5	17.980	27.141	38.042	20.378	4.650	8.320	4.638	1.713	17.419	2.547
PD (Pearson) R	**Antibiotic consumption 2012–2016**			−0.093	−0.228	0.137	0.203	−0.028	−0.124	0.047	−0.026	0.055	0.009
PD (Pearson) *p*				0.624	0.226	0.471	0.281	0.883	0.514	0.805	0.891	0.774	0.964
PD OR				0.995	*0.921*	0.998	1.055	0.941	*0.815*	1.027	1.044	1.066	1.136
PD CI95%				0.862–1.143	*0.833–1.011*	0.932–1.070	0.962–1.164	0.852–1.029	*0.637–1.003*	0.962–1.098	0.952–1.150	0.947–1.208	0.954–1.371
PD *p*				0.940	*0.094*	0.954	0.257	0.190	*0.071*	0.430	0.367	0.297	0.162
PD% (Pearson) R	**Prevalence change of PD 1990–2016 in %**			−0.100	0.255	0.162	−0.062	**0.603**	**0.436**	**−0.405**	**−0.377**	−0.277	**−0.496**
PD% (Pearson) *p*				0.600	0.174	0.394	0.744	**< .001**	**0.016**	**0.026**	**0.040**	0.138	**0.005**
PD% OR				1.009	*1.096*	1.014	0.962	**1.119**	**1.347**	0.941	0.945	0.943	**0.778**
PD% CI95%				0.887–1.147	*0.989–1.219*	0.947–1.088	0.881–1.050	**1.021–1.249**	**1.077–1.779**	0.874–1.011	0.856–1.037	0.839–1.057	**0.641–0.928**
PD% *p*				0.884	*0.082*	0.683	0.378	**0.024**	**0.017**	0.099	0.238	0.312	**0.007**
PD death R	**Death rate attributed to PD/** **100,000 population**			−0.100	0.255	0.162	−0.062	**0.603**	**0.436**	**−0.405**	**−0.377**	−0.277	**−0.496**
PD death *p*				0.600	0.174	0.394	0.744	**< .001**	**0.016**	**0.026**	**0.040**	0.138	**0.005**
PD death OR				1.053	1.029	0.969	0.929	1.032	1.105	0.991	1.007	1.055	0.936
PD death CI95%				0.922–1.214	0.937–1.135	0.899–1.041	0.841–1.017	0.936–1.147	0.888–1.416	0.925–1.061	0.917–1.106	0.929–1.205	0.788–1.103

**Table 5 antibiotics-11-01145-t005:** Comparison of the consumption of narrow-spectrum penicillin (J01CE, J01CF) in countries with the highest prevalence change of PD and the countries with decreased prevalence of PD. The higher consumption of narrow-spectrum penicillin (J01CE, J01CF) promoting the prevalence of PD is well observed in the countries with the top highest prevalence of PD, while, in the countries showing a decreasing prevalence of PD, a reduction in the consumption of narrow-spectrum penicillin (except for Portugal/J01CF/) is seen in the 5-year periods. The difference between the consumption of narrow-spectrum, beta-lactamase sensitive penicillin in countries with a higher increase in PD prevalence is nearly 6-fold compared to countries showing a reduction in the prevalence of PD between 1990–2016, while the difference in the consumption of narrow-spectrum, beta-lactamase resistant penicillin is 3-fold.

**Countries with the Highest Prevalence Change between 1990–2016**	**The Prevalence Change of the PD between 1990–2016 in %**	**Relative Share of Narrow Spectrum,** **Beta-Lactamase Sensitive Penicillin (J01CE) Consumption in %**					**Average**
		**J01CE 1997–2001**	**J01CE 2002–2006**	**J01CE 2007–2011**	**J01CE 2012–2016**		
**Norway**	**93**	30.198	26.052	24.401	21.140		*25.448*
**Denmark**	**56.7**	38.990	37.825	33.496	28.223		*34.633*
**Portugal**	**34.3**	0.227	0.221	0.145	0.081		*0.169*
**UK**	**32.5**	5.009	4.955	4.742	4.650		*4.839*
						**AVERAGE:**	**16.272**
**Countries with the Highest Prevalence Change between 1990–2016**	**The Prevalence Change of the PD between 1990–2016 in %**	**The Relative Share of Narrow Spectrum,** **Beta-Lactamase Resistant Penicillin (J01CF) Consumption in %**					
		**J01CF 1997–2001**	**J01CF 2002–2006**	**J01CF 2007–2011**	**J01CF 2012–2016**		
**Norway**	**93**	1.485	2.896	3.815	3.969		*3.041*
**Denmark**	**56.7**	4.247	6.642	7.362	8.716		*6.742*
**Portugal**	**34.3**	3.246	3.115	2.806	2.633		*2.950*
**UK**	**32.5**	5.141	6.677	7.423	8.320		*6.890*
						**AVERAGE:**	**4.906**
**Countries with a Negative Prevalence Change between 1990–2016**	**The Prevalence Change of the PD between 1990–2016 in %**	**Relative Share of Narrow Spectrum,** **Beta-Lactamase Sensitive Penicillin (J01CE) Consumption in %**					
		**J01CE 1997–2001**	**J01CE 2002–2006**	**J01CE 2007–2011**	**J01CE 2012–2016**		
**Netherlands**	**–6.0**	5.803	4.967	3.952	2.950		*4.418*
**Italy**	**–5.5**	0.233	0.065	0.000	0.000		*0.074*
**Bulgaria**	**–5.3**	15.431	6.841	2.400	1.075		*6.437*
**France**	**–5.1**	0.819	0.745	0.627	0.698		*0.722*
						**AVERAGE:**	**2.913**
**Countries with a Negative Prevalence Changed between 1990–2016**	**The Prevalence Change of the PD between 1990–2016 in %**	**The Relative Share of Narrow Spectrum,** **Beta-Lactamase Resistant Penicillin (J01CF) Consumption in %**					
		**J01CF 1997–2001**	**J01CF 2002–2006**	**J01CF 2007–2011**	**J01CF 2012–2016**		
**Netherlands**	**–6.0**	2.648	2.896	3.653	4.498		*3.424*
**Italy**	**–5.5**	0.124	0.092	0.043	0.044		*0.076*
**Bulgaria**	**–5.3**	0.397	0.125	0.000	0.000		*0.130*
**France**	**–5.1**	1.910	1.899	1.719	1.001		*1.632*
						**AVERAGE:**	**1.316**

## Data Availability

Not applicable.

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
