# Peer review of "Antibiotic Consumption Patterns in European Countries Are Associated with the Prevalence of Parkinson’s Disease; the Possible Augmenting Role of the Narrow-Spectrum Penicillin"

_antibiotics, 2022, doi:10.3390/antibiotics11091145_

Round 1

Reviewer 1 Report

Dear Authors,

It is a very interesting subject in terms of approach and presentation. The correlation between the occurrence of neurodegenerative pathology and in this case Parkinson's disease, with the gut microbiota alteration, is an interesting approach that opens alternatives for elucidating some mechanisms of the occurrence of these neurodegenerative diseases.

From begin, in my opinion, the title limits the scope of addressing the antibiotics taken in this study, because the statistics presented in the 5 tables include several classes of antibiotics, some of which increase and others decrease the incidence of PD symptoms, as can be seen from the discussion chapter. I suggest the authors to reformulate the title in accordance with the purpose of this study.

In order to have a clearer picture, I suggest a few observations to the authors:

- Line 23: In the abstract, the recent research on PD is highlighted, and the reference is to the year 2016. I suggest that it be reformulated considering that there is much more recent research on this subject, even from 2021.

- Line 41: authors should add more recent bibliographic index - for example WHO statistics 1990-2019.

-Lines 56-64: there are recent studies on the causes of the occurrence of PD - I suggest that the authors include among the determining factors in the occurrence of PD genetic mutations with the repetition of the triplet of CAG nucleotides in the so-called polyglutamic diseases, which also include PD.

- Line 82: For a better understanding of the degenerative pathological process in PD, I suggest bringing additional explanations about the connection between PD and gut-microbiota changes, highlighting the importance of SCFA (short-chain fatty acids) in this process.

- Tables should have the title above the table, not after the table.

- Line 183: it is stated that there are many studies in relation to the abundance and diversity of microbial taxa, and yet to give only one bibliographic index. Please give examples of other studies that highlight this aspect.

Author Response

To Rev. No. 1.

Thank you for your observations and kind advice

  1. From begin, in my opinion, the title limits the scope of addressing the antibiotics taken in this study, because the statistics presented in the 5 tables include several classes of antibiotics, some of which increase and others decrease the incidence of PD symptoms, as can be seen from the discussion chapter. I suggest the authors reformulate the title by the purpose of this study.

Answer: The title had been reformulated (yellow color)

  1. Line 23: In the abstract, the recent research on PD is highlighted, and the reference is to the year 2016. I suggest that it be reformulated considering that there is much more recent research on this subject, even from 2021.

Answer: The PD database standing for 2016 was published in 2018. This database is very detailed and comprehensive, and similar databases were not available, even in the WHO reports, but the latest WHO summary on PD statistics (which stands for 2019) was included (yellow marking color) as:

As of WHO report (13 June 2022), globally, disability and death due to PD are increasing faster than for any other neurological disorder. The prevalence of PD has doubled in the past 25 years. Global estimates in 2019 showed over 8.5 million individuals with PD. Current estimates suggest that, in 2019, PD resulted in 5.8 million disability-adjusted life years, an increase of 81% since 2000, and caused 329 000 deaths, an increase of over 100% since 2000 (https://www.who.int/news-room/fact-sheets/detail/parkinson-disease).

  1. Line 41: authors should add more recent bibliographic index - for example WHO statistics 1990-2019.

Answer: Added as above

  1. Lines 56-64: there are recent studies on the causes of the occurrence of PD - I suggest that the authors include among the determining factors in the occurrence of PD genetic mutations with the repetition of the triplet of CAG nucleotides in the so-called polyglutamic diseases, which also include PD.

Answer: Possible causative agents were referred along with the genetic background (Ref. 11), but the basic pathology had been discovered earlier (Lewy bodies, Braak theory, etc.), which is reflected in the references. Out of the 42 references 27 were published in the past 5 years, and 10 references were published in 2020-21-22.

  1. Line 82: For a better understanding of the degenerative pathological process in PD, I suggest bringing additional explanations about the connection between PD and gut-microbiota changes, highlighting the importance of SCFA (short-chain fatty acids) in this process.

Answer: The possible role of reduced SCFA, related to Parkinson’s related dysbiosis is mentioned at the end of the article. It might be suspected also that the altered microbiome produces less SCFA (short-chained fatty acids), as SCFA induces regulatory T (Treg) cells, a decrease of SCFA-producing bacteria may be a prerequisite for the development of PD.

  1. Tables should have the title above the table, not after the table.

Answer: The legend of the tables were placed above the tables

  1. Line 183: it is stated that there are many studies in relation to the abundance and diversity of microbial taxa, and yet to give only one bibliographic index. Please give examples of other studies that highlight this aspect.

Answer: This single reference (20) is a summary of several publications related to the issue.

Reviewer 2 Report

I have evaluated the manuscript (Antibiotics-1845038) titled “Narrow-spectrum penicillin consumption might be associated with the prevalence of Parkinson’s disease in European countries” by Ternák and coworkers, about the prevalence of Parkinson’s disease in European countries which could be due to penicillin consumption. All standard methods were used for this research. I found this article interesting for the readers and followed the journal Antibiotics’ scope.

I would recommend the article could be published in Antibiotics after minor corrections.

1.     According to recent research in 2016”: The author should comment on why it took so long to write the manuscript although the data are before 2016.

2.     Introduction needs to be concise.

3.     The author needs to elaborate a bit in the conclusion (missing) to show the future direction of this study.

4.     Tables 1 to 5, need heading and footnotes.

5.     The author could highlight the data in the table in the discussion section.

 6.     The authors need to follow the journal’s standard reference format available in the portal.

Author Response

Thank you for your observations and advices.

  1. “According to recent research in 2016”: The author should comment on why it took so long to write the manuscript although the data are before 2016.

Answer: The PD database standing for 2016 was published in 2018. This database is very detailed and comprehensive, and similar databases were not available, even in the WHO reports, but the latest WHO summary on PD statistics (which stands for 2019 !) was included (yellow marking color) as:

As of WHO report (13 June 2022), globally, disability and death due to PD are increasing faster than for any other neurological disorder. The prevalence of PD has doubled in the past 25 years. Global estimates in 2019 showed over 8.5 million individuals with PD. Current estimates suggest that, in 2019, PD resulted in 5.8 million disability-adjusted life years, an increase of 81% since 2000, and caused 329 000 deaths, an increase of over 100% since 2000 (https://www.who.int/news-room/fact-sheets/detail/parkinson-disease).

  1. Introduction needs to be concise.

Answer: Corrected

  1. The author needs to elaborate a bit in the conclusion (missing) to show the future direction of this study.

Answer: Conclusion had been separated from the Discussion and the following sentences were added: Our observation indicated the possible role of antibiotic consumption in the development of PD and our results are further strengthened by the fact that countries with high consumption of narrow-spectrum penicillin have experienced higher prevalence increase of PD, while others, when reducing or ceasing the consumption of narrow spectrum penicillin, experienced a reduction in the prevalence of PD in the previous 25 years. Further studies might elucidate the molecular background of this mechanism.

  1. Tables 1 to 5, need heading and footnotes.

Answer: The legends of the tables had been placed above the tables.

  1. The author could highlight the data in the table in the discussion section.

Answer: Our results (included in tables) had been highlighted and explained in the Discussion and Conclusion, but some explanatory sentences were added also as: Our observation indicated a positive statistical association and higher risk for the development of PD in countries consuming higher amount of narrow spectrum penicillin, which is demonstrated in Fig. 2, showing the association between the PD prevalence increase and the consumption of narrow spectrum penicillin for the time period 1997-2001. Similar associations had been detected in the other time periods as well, but statistically significant negative associations and reduced risk of PD has been detected on comparing the consumption of broad spectrum antibiotics (quinolone) and the PD prevalence also. The opposing effect of the consumption of “promoter” (penicillin) and “inhibitor” antibiotics (broad spectrum antibiotics) might determine the actual prevalence of PD along with other factors playing role in the development of PD (genetics).

  1. The authors need to follow the journal’s standard reference format available in the portal.

Answer: Corrected

Reviewer 3 Report

Dear Authors,

The study you presented is very interesting and original. I rate it very high, and I have only two remarks that I think may improve the paper:

1. Please separate a paragraph at the end of the discussion that would act as a summary or short conclusions section to pinpoint the main findings of your study.

2. Do consider including a graphical presentation of your main observations (e.g. a graph summarizing the number of PD incidents and the use of antibiotics in the selected countries).

Author Response

To the Reviewer No 3.

Thank you very much for your kind observation and advice. The following corrections were performed:

  1. Please separate a paragraph at the end of the discussion that would act as a summary or short conclusions section to pinpoint the main findings of your study.

Answer: Separation of Discussion and Conclusion was introduced

  1. Do consider including a graphical presentation of your main observations (e.g. a graph summarizing the number of PD incidents and the use of antibiotics in the selected countries).

Answer: 2 graphic figures were included